# Study of the Fatty Acid Profile of Milk in Different Sheep Breeds: Evaluation by Multivariate Factorial Analysis

**DOI:** 10.3390/ani12060722

**Published:** 2022-03-13

**Authors:** Giuseppe Conte, Valentino Palombo, Andrea Serra, Fabio Correddu, Mariasilvia D’Andrea, Nicolò Pietro Paolo Macciotta, Marcello Mele

**Affiliations:** 1Department of Agriculture, Food and Environment, University of Pisa, Via del Borghetto 80, 56124 Pisa, Italy; andrea.serra@unipi.it (A.S.); marcello.mele@unipi.it (M.M.); 2Research Center of Nutraceutical and Food for Health, University of Pisa, Via del Borghetto 80, 56124 Pisa, Italy; 3Dipartimento Agricoltura, Ambiente e Alimenti, Università degli Studi del Molise, Via De Sanctis snc, 86100 Campobasso, Italy; abg@unimol.it (V.P.); dandrea@unimol.it (M.D.); 4Department of Agriculture, University of Sassari, Via de Nicola 9, 07100 Sassari, Italy; fcorreddu@uniss.it (F.C.); macciott@uniss.it (N.P.P.M.)

**Keywords:** sheep, breed, factor analysis, fatty acids, milk fat

## Abstract

**Simple Summary:**

The quality of milk is strongly influenced by its lipid profile. The increase in fats with nutraceutical properties at the expense of those negative for human health, has always been a goal to improve the functional properties of milk. To achieve this goal, it is essential to know the metabolism of the mammary gland and the relationship between the various lipid components. Much is known about bovine milk, while the aspect relating to the sheep species has not been developed. The present work aims to investigate the relationships between the various fatty acids in sheep’s milk through a multivariate approach, which can highlight the mammary role of lipid synthesis.

**Abstract:**

A multivariate analysis was used to investigate the fatty acid (FA) profile in three different Italian sheep breeds: Comisana, Massese, and Sarda. A sample of 852 animals was considered: 118 Massese, 303 Comisana, 431 Sarda. Sarda sheep were divided into two groups, based on their breeding origin (298 and 133 reared in Sardinia and Tuscany, respectively). Sarda sheep, bred both in Sardinia and in Tuscany, were considered in different groups, both because in these two regions most of the sheep of this breed are reared, and because they differ in geographical characteristics and in the farming system. The individual milk FA composition of dairy ewes was analyzed with multivariate factor analysis. The extracted factors were representative of the following eight groups of fatty acids or functions: factor 1 (odd branched fatty acids and long-chain fatty acids), factor 2 (sn3_position), factor 3 (alternative biohydrogenation), factor 4 (SCD_1), factor 5 (SCD_2), factor 6 (SCD_3), factor 7 (fat secretion) and factor 8 (omega-3). A factor analysis suggested the presence of different metabolic pathways for de novo short- and medium-chain fatty acids and Δ9-desaturase products. The ANOVA of factor scores highlighted the significant effects of the breed. The results of the present study showed that breed is an important factor in defining the fatty acid profile of milk, combined with the effect of the diet. Breeds reared in the same farming system (Comisana, Massese and Sarda reared in Tuscany) showed significant differences for all the factors extracted. At the same time, we found differences between the Sarda sheep reared in Sardinia and Tuscany, two different regions of Italy.

## 1. Introduction

The milk production of small ruminants, particularly sheep, represents a sustainable source for many countries, especially for those areas where the arid climate and adverse environmental conditions constrain the use of specialized dairy breeds. Ewe farming systems are commonly characterized by the use of autochthonous breeds, which allow genetic variability to be preserved and minimize costs [1].

The fatty acid composition of sheep milk is characterized by a high level of some short-chain fatty acids (SCFA), particularly capric acid (C10:0) [2]. Furthermore, compared to cattle, the fatty acid (FA) profile of sheep milk shows substantial differences, most likely related to the different regulation of some pathways of lipid metabolism, as is the case with the elongation processes of FA, which are synthesized de novo [2]. Similar results were reported for the comparison between sheep and goat. Nudda et al. [3] showed that conjugated linoleic acid isomers (CLA) in milk were higher in sheep than in goat when diet was mainly based on pasture, while no differences were observed when the supply of concentrate was increased.

Although sheep milk is almost entirely used for cheese production [4], dairy sheep breeding programs have historically aimed to improve total milk yield per lactation [5], and the selection for milk composition is carried out only in few breeds [6,7]. This is mainly due to the high recording costs compared to income per ewe [8,9]. On the other hand, the growing consumer interest in the nutritional quality of dairy products pushes toward the inclusion of fine milk composition traits in the breeding goals of dairy species. Although animal feeding is considered to be the most important factor affecting milk FA composition [10,11], the genetic variation of these traits has been reported in cattle [12,13] and sheep [11,14], suggesting the possibility of a genetic improvement.

Genomic studies on milk FA in cattle have mainly focused on the evaluation of their genetic determinism [15,16,17]. In dairy sheep, the molecular basis of FA has been investigated by a candidate gene [18,19], QTL detection [8], and genomic [20] approaches. However, the interpretation of FA variability in sheep is relatively more complex than in cattle due to the strong effect of management and environmental factors and the reduced selection pressure to which this species is subjected. Hence, information regarding the metabolic status of the ewes could be obtained by evaluating the simultaneous variations in the FA groups rather than considering individual FA.

A multivariate statistics approach is able to summarize the information contained in a complex system defined by several variables with a smaller number of new explanatory variables, allowing for an easier interpretation of the original multivariate system. A suitable technique for studying the (co)variance system of the milk FA profile is a multivariate factor analysis (MFA). The MFA has previously been used to analyze milk FA composition in dairy cows [21,22,23], buffalo [24] and sheep [25,26]. The results of these studies confirmed that the usefulness of MFA was a helpful method for analyzing the complex pattern of correlations among FA by the generation of a few uncorrelated synthetic variables with clear technical and biological meanings to be used as new phenotypes in further analyses.

In the present study, MFA was used to analyze the detailed milk FA compositions of a sample of dairy sheep from three Italian breeds (Massese, Comisana and Sarda). Our aims were (1) to study the correlation patterns among FA in sheep milk, and (2) to derive new synthetic variables to explain the mammary metabolism in order to generate new parameters for assessing differences among breeds in the FA profile.

## 2. Materials and Methods

### 2.1. Animals, Breeds and Dairy Systems

A sample of 852 animals from three Italian dairy breeds was considered: 118 Massese (Mas) from 3 farms (39 ± 7 sheep per farm), 303 Comisana (Com) from 4 farms (76 ± 30 sheep per farm), and 431 Sarda. The animals of the Mas and Com breeds were all reared in farms of Tuscany (a region of Central Italy), while Sarda sheep were farmed partly (*n*° = 298, from 4 farms—76 ± 25 sheep per farm) in Sardinia (one of the two major Italian islands) (Ss) and partly (*n*° = 133, from 3 farms—44 ± 5 sheep per farm) in Tuscany (St). Sarda sheep were divided up on the basis of their geographical origin, because Tuscany and Sardinia are the two Italian regions where most of Sarda sheep are farmed, and because they differ in geographical characteristics and in farming systems (Appendix A). On average, 30 ± 15.9 sheep per flock were considered. All of the sheep were pluriparous and were in the same mid-lactation stage (100 ± 10 days in milking) and milk production level (1500 ± 100 mL/day). In general, the flocks were selected from farms homogeneous for feeding and farming systems, with animals brought to pasture in the spring and summer months and kept in the stable during the winter seasons when they were fed on meadow hay, mainly produced on the farm, and a small–medium amount of compound feed from the feed industry. The main differences between the sheep were due to the breeding region (Sardinia or Tuscany), because the pasture was based on both spontaneous and artificial pastures that are typical of the geographical region. Since Mas, Com and St are reared in Tuscany, they find themselves in the same feeding and farming system conditions. However, for all sheep, the main dry matter intake comes from grazing, while the added consumption of hay takes place in sheepfold and from concentrates (approx. 800 g d^−1^), normally in the milking parlor. Milk samples (one per ewe) were collected during the morning milking. The milk samples (no preservative was added) were immediately refrigerated at 4 °C and transferred to the Laboratory of the Department of Agriculture, Food and Environment of the University of Pisa for Mas, Com and St samples, and Department of Agriculture of University of Sassari for Ss samples.

### 2.2. Fatty Acid Composition

Milk fat extraction and the derivatization of FA were carried out following the procedure described by Mele et al. [27]. Milk FA composition was determined by gas chromatography (GC) analysis using a GC2010 Shimadzu gas chromatograph (Shimadzu, Columbia, MD, USA) equipped with a flame ionization detector and a highly polar fused-silica capillary column (Chrompack CP-Sil88 Varian, Middelburg, The Netherlands; 100 m, 0.25 mm i.d.; film thickness 0.20 mm). Hydrogen was used as the carrier gas at a flow of 1 mL/min. Split/splitless injector was used with a split ratio of 1:80. An aliquot of the sample was injected under the following GC conditions: the oven temperature started at 40 °C and was maintained at that level for 1 min, then increased to 173 °C at a rate of 2 °C/min, and was maintained at that level for 30 min. It was then, once again, increased to 185 °C at 1 °C/min and maintained for 5 min, and finally to 220 °C at a rate of 3 °C/min, and held for 19 min. The injector temperature was set at 270 °C, and the detector temperature was set at 300 °C. Individual FA methyl esters (FAME) were identified by comparison with a standard mixture of 52 Component FAME Mix (Nu- Chek Prep Inc., Elysian, MN, USA). The identification of isomers of C18:1 was based on commercial standard mixtures (Supelco, Bellefonte, PA, USA) and published isomeric profiles [28]. Nonanoic and nonadecanoic methyl esters were used as internal standards for short- and medium–long-chain FA, respectively. Milk FA composition was expressed as grams per 100 g of total lipids (TL). A total of 33 FAs were analyzed in this study.

### 2.3. Statistical Analysis

Factor analysis. The objective of MFA is to describe the (co)variance of a system defined by *n* traits (y1, …, yn), measured on observation units by deriving a smaller number *p* (*p* < *n*) of latent unobservable variables (X1, …, Xp), named common latent factors. Factor analysis assumes that the variance of each original variable can be decomposed into two components, one common to all variables and one specific for each variable, named as communality and uniqueness, respectively. The factor model decomposes the covariance matrix of the measured traits (S) as follows:S = BB′ + Ψ 
where BB′ and Ψ are the communality and the uniqueness (co)variance matrices, respectively [29]. According to the (co)variance model, the measured traits can be represented as a combination of ***p*** unobservable common factors (X) plus a unique variable (e):y_1_ = b_11_X_l_ + … + b_1p_X_p_ + e_1_

y_n_ = b_n1_X_l_ + … + b_np_X_p_ + e_n_

where X_j_ is the jth common factor, b_ij_ are factor coefficients (or loadings, i.e., correlations between the jth common factor and the ith trait) [29]. Loadings are the elements of the B matrix used in factor model. Common factors create covariances between original variables, whereas the residual specifically contributes only to the individual variation. The MFA was carried out on the correlation matrix of 33 FA measured in the 852 ewes using JMP software of SAS (SAS Inst. Inc., Cary, NC, USA).

In order to test the adequacy of data sets used for the factor analysis, the Kaiser Measure of Sampling Adequacy (Kaiser MSA) was calculated. This parameter summarizes the difference between Pearson and partial correlations [30]. The number of factors to be extracted was based on their eigenvalues (>1), their readability in terms of relationships with the original variables, and the amount of explained variance. Factor readability was improved through a VARIMAX rotation. A variable was considered as related to a specific factor if the absolute value of its loading was ≥ 0.60 [31].

Factor scores were calculated for each ewe according to the following formula:x′ = y′ × (BB′ + Ψ)^−1^ × B, 
where x′ is the row vector of factor scores, y′ is the row vector of standardized (value − mean)/standard deviation) traits. Standardized values, instead of raw values, were used because analyzed traits had different units of measurement and scale.

Univariate analysis. Individual factor scores were then used as new phenotypes and analyzed with the following mixed linear model:y_ijz_ = μ + breed_i_ + flock_j_[breed_i_] + ε_ijz_

where y_ijz_ is the factor score (fatty acids and cholesterol); μ is the overall mean; breed_i_ is the fixed effect of the ith breed (i = Mas, Com, St and Ss); flock_j_ = is the random effect of the jth flock (1 to 28); and ε_ijz_ is the random residual term.

## 3. Results and Discussion

### 3.1. Milk FA Composition

Data relative to descriptive statistics of the milk FA composition are reported in Table 1 (descriptive statistics for each breed are reported in Appendix A). The FA profile comprised 15 SFA, 11 monounsaturated FA (MUFA) and 7 PUFA (PUFA). The most abundant FA were palmitic acid (C16:0), oleic acid (C18:1c9), myristic acid (C14:0) and stearic acid (C18:0), which together represented 52% of the total FA. All FA showed high variability, with the coefficient of variation ranging from 14.78% for C16:0 to 66.26% for C22:6 n-3 (Table 1). As a whole, PUFA and, in particular, those with C > 20 showed the highest variation. The four groups of sheep showed the same level of fat percentage: Ss 6.04 ± 0.94, St 5.75 ± 0.73, Com 6.22 ± 1.13, Mas 6.11 ± 0.85.

### 3.2. Multivariate Factor Analysis

Numerous studies have pointed out the difficulty of assessing the relationships between several variables used to describe the nutritional and technological quality of milk (fatty acid profile, milk coagulation properties, protein composition etc.) [21,32,33]. This condition represents a seriously limited large-scale implementation of selection and management strategies that aim to improve milk technological quality. Moreover, it was revealed that sampling errors help to increase the interpretation of results, especially when many traits are evaluated [34]. Therefore, the application of MFA may be very useful for reducing the complexity of the system, as demonstrated in previous studies of cow [21,22,23] and buffalo milk [24]. Few uncorrelated variables may be consistent markers to described mammary metabolism and define milk quality. The decision to only collect milk samples in spring is related to the aim to standardize the four groups as much as possible. As is well-known, the FA profile depends on several factors (animal, diet, environment, lactation stage, parity). To evaluate the mammary metabolism according to the breed, we preferred to standardize all the other factors, including the season. Clearly, the season especially has an effect on those factors related to diet. However, from our previous experiences on cattle [22], we found that the effect of the season on factors plays a marginal role. The Kaiser MSA was 0.81, higher than 0.80, which is considered an empirical threshold that flags a dataset as suitable for MFA [35]. In fact, a value higher than 0.80 means that the partial correlations values were significantly lower than Pearson correlations, signifying that the association between two variables was regulated by other variables present in the dataset. So, it is possible to conclude that a latent correlation structure exists.

Eight latent factors, able to explain approximately 79% of the total variance, were extracted by MFA from the FA correlation matrix (Table 2). The explained variance was efficiently partitioned among the factors, with factor 1 showing a small predominance (eigenvalue 7.32), whereas the eigenvalues of the other 7 factors ranged between 1.22 and 5.76 (Table 2). This is a typical characteristic of MFA in comparison with principal components analysis, where the first component is usually related to a larger amount of variance than successive variables [36]. The pattern of explained variance among extracted factors is characteristic of MFA as demonstrated in previous studies [21,22,23,24].

In total, 27 out of 33 FA showed loading values of >0.60 with only one factor. This result highlights a simple structure, which represents an indicator for the suitability of the factor model assumption for the analyzed data [29]. An exception is represented by C22:5n3 and C22:6n3, which exhibited loadings >0.60 for two factors.

The first extracted factor (Factor 1) was named “OBCFA and LCFA”, as it was positively correlated with branched-chain FA (BCFA), odd-chain FA (OCFA) and long-chain FA (LCFA, C ≥ 20). BCFA and OCFA in milk originate mainly from bacterial flora present in the rumen. In particular, these FA are synthesized and used by rumen bacteria to regulate the optimal fluidity of the microbial cell membrane [37]. Since the growth and activity of ruminal microorganisms are affected by diet characteristics, the concentration and the relative abundance of BCFA and OCFA in milk are affected by the diet [38]. Therefore, BCFA and OCFA concentrations in milk could be used as investigative tools to predict shifts in microbial population, principally related to the variation of diet composition [39,40]. Moreover, some LCFA (C20:0, C20:4n6, C24:0, C22:5n3 and C22:6n3) were also positively associated with this factor. These FA may derive either from diet and fatty acid elongation and desaturation. In previous works LCFA were always grouped into a separate factor, both in dairy cows [21,22] and buffaloes [24]. In sheep, we observed a close relationship with OBCFA. This result suggests a possible metabolic pathway that may be representative of the mammary metabolism of sheep and not of other ruminants. OBCFAs and LCFAs present in milk are imported into mammary epithelia cells from the plasma after being either released from triglycerides circulating in chylomicra or very low density lipoprotein by the enzyme lipoprotein lipase (LPL) [41]. This enzyme is generated in the epithelial cells of the mammary gland and regulates the flow of FA within the epithelial cells [41]. Crisà et al. [18] revealed that LPL influences the PUFA levels in sheep milk, confirming the possible role of this gene in factor 1 definition.

Factor 2 was positively correlated with short- and medium-chain FA, with the exception of C4:0 and C6:0, and negatively correlated with C18:1c9 (Table 2). These FA are de novo synthesized in the mammary gland from acetate by the acetyl CoA carboxylase (ACC) and FA synthase (FAS) enzymes [42]. Moreover, C18:1c9 is related to the activity of the stearoyl-CoA desaturase (SCD), which catalyzes the desaturation of the C18:0 at the Δ9 position.

Some recent studies on dairy cows and buffaloes found a factor with a similar loading structure [21,22,23,24]. The second latent factor was, therefore, associated with the mammary gland activity and, in particular, with the regulation of milk fat fluidity. De novo short-chain FA (from 4 to 10 carbons) and C18:1c9 are preferentially esterified at position sn3 of glycerol, playing a crucial role in the regulation of milk fat fluidity [42]. The opposite loadings observed in factor 2, between de novo FA and C18:1c9, were in agreement with previous reports on dairy species [21,22,23,24], emphasizing that this regulation system is typical for all ruminants. Timmen and Patton [43] proposed that the increase in milk C18:1c9 due to the activity of SCD on C18:0 could be considered as a mechanism of milk fat fluidity maintenance when availability of de novo FA is reduced. For these reasons the second factor was named “sn3_position”.

The third latent factor was positively related to some intermediate products of rumen biohydrogenation (C18:1c12, C18:1t6–8, C18:1t9, C18:1t10, and C18:1t16) (Table 2). Vaccenic acid (C18:1 t11), the main product of this pathway, did not have a large loading on this factor, and it was instead included in factor 5. Linoleic (C18:2n6) and α-linolenic (C18:3n3) acid are often the main FA contained in dietary lipids and are actively biohydrogenated by rumen bacteria to stearic acid (C18:0) [44]. Similar results were observed by Conte et al. [21] in cattle and by Correddu et al. [24] in buffalo. The FA associated with this factor are linked by a common metabolic origin, being the intermediates of ruminal biohydrogenation of long-chain PUFA. Usually, the main biohydrogenation pathway consists of a reduction of dietary PUFA to C18:0 via C18:1t11 [44]. The decrease in rumen pH often results in bacterial population shifts and consequent changes in the pattern of fermentation end products [45]. Leat et al. [46] reported that changes in rumen bacteria populations are associated with modifications in the biohydrogenation pathways consistent with the altered trans-octadecenoic acid profile found in ruminal digesta and tissue lipids. In addition, Griinari et al. [47] revealed that an alteration of rumen environment induced by feeding high-concentrate diets is related with a modification in the trans-octadecenoic acid profile of milk fat. During this situation, C18:1t10 replaced C18:1t11 as the predominant trans C18:1 isomer in milk fat. Pathways for the production of C18:1t10 were hypothesized [48], and these include a specific cis-9, trans-10 isomerase in rumen bacteria with the synthesis of CLAt10c12 as the first intermediate. For this reason, the third factor was then named “alternative biohydrogenation”.

The fourth latent factor was positively associated with the C10:1c9 and C14:1c9 (Table 2). These MUFA are two of the main products of the SCD activity on their correspondent saturated substrates. So, this factor was interpreted as an index of the SCD activity and then was named “SCD_1”. In fact, other products of SCD activity (C16:1c9, C18:1c9 and C18:2c9t11) were associated with other factors. This is not surprising as these FA are both involved in different metabolic pathways, such as ruminal biohydrogenation, mammary gland desaturase activity and milk fat fluidity regulation. Furthermore, our findings were in agreement with previous reports in dairy cattle [21,22,24].

Factor 5 was positively associated with C18:1t11 and C18:2c9t11, which are positively associated with the fifth factor. For this reason, the factor was named “SCD_2”. As previously described, vaccenic acid is the main intermediate fatty acid of ruminal biohydrogenation. A quote of vaccenic acid may bypass the rumen environment and accumulate in the mammary gland where it is partially converted to C18:2c9t11 by the SCD [49]. The results of the present study are in agreement with a previous report on dairy cattle [21,22] and buffalo [24] where a latent factor highly correlated with C18:1t11 and C18:2c9t11 was found.

The sixth factor was positively associated with C16:0 and C16:1c9, which represent another substrate/product pair of SCD activity. For this reason, the sixth factor was named “SCD_3”. This factor was not observed in the previous MFA in the composition of milk FA from dairy cows and buffalo. Palmitic acid (C16:0) represents the final product of milk FA synthesis. As demonstrated in previous studies [21,22,24], SCD activity may be split into different factors (SCD_1, SCD_2 and SCD_3), confirming the role of this gene in the regulation of mammary lipid metabolism in different lipid metabolism pathway. All three factors summarized the relationship between substrates and related products of the SCD enzyme. Since MFA extracts factors that are independent from each other, this leads us to believe that the pathways expressed by the SCD are independent of each other and therefore we have considered them as three regulatory systems of the SCD enzyme. Evidently, further investigations are needed to demonstrate this role.

The seventh latent factor was named “fat secretion” because it was positively correlated with the contents of C18:0 and negatively with C4:0 and C6:0. As observed in previous studies in dairy cows [21,22,23] and buffalo [24], C4:0 and C6:0 were associated with a different factor than the other short- and medium-chain SFA (C8:0 to C14:0), although they are all endogenously synthesized in the mammary gland by ACC and FAS enzymes [42]. This result further confirms that differences may be present in the endogenous synthesis of even-chain FA according to the carbon chain length. Contrary to medium-chain FA (such as from C8:0 to 14:0), short-chain FA may be partly synthesized in the mammary gland by metabolic pathways not dependent on ACC [50]. From a factor analysis, we were able to highlight this metabolic difference by extracting two different latent variables, one representing short-chain and one representing medium-chain FA metabolism. Unlike other studies of ruminants, it is observed that this factor is associated with C18:0 even if it has the opposite sign to C4:0 and C6:0. The C18:0 cannot be synthesized by the mammary gland, and it is derived from the bloodstream through different sources: (1) dietary FA as affected by ruminal biohydrogenation [37] and (2) the mobilization of lipid deposits [50]. This factor explains the pathway that regulates the lipid source mainly used by the mammary gland for the synthesis of milk fat. As is known from the literature, C18:0 and C18:1c9 are the most representative FA that are not of mammary origin [50,51]. C18:0 and mammary de novo synthesis FA were suggested to have a complementary role in lipid and energetic metabolism in dairy cows [52]. Several works demonstrated that mammary de novo synthesis FA and C18:0 had opposite trends in their relationship to energy balance [15,53,54]. In the first phase of lactation, when the animal usually experiences a negative energy balance, the mammary de novo synthesis of FA is reduced, and milk FA are mainly derived from extramammary sources. In this case, the relative abundance of circulating C18:0 is higher [53]. Moreover, recent research suggested that mammary de novo synthesis FA and C18:0 are important regulators of metabolism and gene transcription in ruminants [55]. This may be an adaptive mechanism for ruminants to regulate metabolism in response to changes in the availability of the more prevalent SFA. Crisà et al. [18] revealed a direct regulation of GHR and the level of C4:0 and C18:0 in ovine mammary gland. The incorporation of C4:0 in milk fat increases with C16:0, which is negatively correlated with C18:0 [51]. In a previous work, an MFA on milk FA from dairy cows extracted a similar factor positively and negatively associated with C16:0 and C18:0, respectively [56].

Factor 8 was positively associated with C20:5n3 (eicosapentaenoic acid, EPA), C22:5n3 (docosapentaenoic acid, DPA) and C22:6n3 (docosahexaenoic acid, DHA) (Table 2). A similar factor was extracted by Correddu et al. [24] in buffalo, while in dairy cattle, omega-3 is associated with omega-6 in the same factor [21,22]. The results of the present work suggest that ewes particularly promote the elongation of C18:3n3, since they are reared in extensive systems with a low input and high availability of pastures. An investigation of milk FA composition from sheep fed diets rich in C18:2n6 or C18:3n3, using a principal component analysis, highlighted an opposite sign of eigenvector coefficients for PUFAn6 and PUFAn3 in the same principal component that was named “n–6 to n–3 ratio” [26]. Similarly, the use of the C18:2n6 to C18:3n3 ratio proved to be very effective in the differentiation between dairy goats fed diets supplemented with different lipid sources [57].

Finally, C18:2n6, C18:3n3, C20:1c11 and C21:0 were not associated with any factor and presented values of communality lower than 0.50. According to the MFA theory, when a variable presents small values of communality (less than 0.4), the descriptive power of the variable might be better represented by the individual variable [21]. So, on the basis of the pattern of factor analysis, these FA were uncorrelated with the other variables, and they seemed to be excluded by the metabolic patterns associated with the eight factors extracted. C18:2n6 and C18:3n3, in particular, represent the principal FA in the feeding regimen and they are not involved in the lipid metabolism of the animal [58]. Therefore, their exclusion was probably related to their scarce association with mammary gland. A similar result was observed by Conte et al. [21] for C18:3n3 in dairy cows. On the contrary, we have little information about C21:0 and C20:1c11, and so we cannot explain their small communality value.

### 3.3. Effect of Breed on the Extracted Factor Scores

The literature on the relationship between breed and milk FA profile in dairy sheep is rather scarce, likely because each breed, and particularly local breeds, are reared in a very small geographical area [1]. In general, the breed effect on milk FA in sheep is of a lower magnitude compared with the diet [59,60,61]. In the case of the present study, a significant (*p* < 0.001) effect of the breed was observed for all the extracted factors, thus highlighting a different specialization about mammary lipid metabolism. Least squares means of factor scores for the three breeds are reported in Figure 1.

Comisana was the only breed that showed positive mean scores for factor 1; Mas and St showed significantly higher largest absolute values, whereas Ss scores were close to 0. As previously stated, this factor is associated with OBCFA, which are considered to be biomarkers of rumen activity, being ruminal bacteria population affected by diet. However, it must be emphasized that the Com, Mas and St breeds show different scores for factor 1, even if they are reared in a similar feeding system. Thus, the difference observed suggests that OBCFA contents were affected by breed as reported in Hanus et al. [62] and Bainbridge et al. [63] for dairy cows. Several studies revealed large variations in the milk FA content and highlight the main role of nutrition in changing the FA profile in different dairy animals [10,42,64]. On the contrary, few studies are available on the genetic determinism of milk FA traits [65,66]. However, the results of this work show that the genetic component plays a non-secondary role in defining the FA profile of milk. The breeds involved showed that there is a different evolutionary strategy in both mammary and ruminal lipid metabolism. In the latter case it is possible to hypothesize a different co-evolution of the breeds with the rumen flora, which leads to differences in the evolution of the metabolic process [37]. So, sheep breeds can be considered as a good model species to study mammary metabolic variation. Contrary to what is observed in species where intense artificial insemination is applied, sheep are characterized by a large within-breed heterogeneity [67,68]. This variability, improved by the wide range of environments and farming systems, results in the existence of different sub-populations within the same breed [20]. This demonstrates what was observed in the present study for the Sardinian breed of sheep reared in two different Italian regions.

Positive mean scores for factor 2 and factor 6 were observed for Mas and Ss, which were higher for the latter breed, whereas Com and St showed statistically similar negative scores. These results highlight a possible greater predisposition of Ss and Mas in mammary lipid neosynthesis, as opposed to St and Com. Mammary lipid neosynthesis is directly influenced by the lactation stage of the animal [69]. In our study, the animals were all in the middle of lactation; therefore, we supposed that the differences observed could be attributed to the different specialization of the breed.

The St was the only breed that exhibited positive mean scores for factor 3, Mas showed negative values, whereas Com and Ss showed intermediate values close to 0. This factor explains the alternative biohydrogenation pathway, which is affected by diet and farm systems [44]. As previously discussed for factor 1, Mas, Com and St were reared in a similar farming system; therefore, from factor 3, it can be hypothesized that a coevolution of the breeds with the rumen bacteria were involved in the biohydrogenation pathway. Similar results were observed by Daghio et al. [70] in two bovine breeds; however, further investigations are needed to demonstrate this effect.

The values of factor 4 were significantly lower for Ss, being the only breed with negative mean scores. The other breeds showed positive values, with the Mas having the largest means. Factor 5 showed significantly lower scores for Ss; the other breeds were characterized by positive values, with Mas having significantly higher scores than the others. The Ss breed had significantly higher scores for factor 7, whereas the Com breed had even negative values. On the contrary, Mas and St presented intermediate values close to 0. Finally, factor 8 showed positive scores for Com and Mas, whereas the two groups of the Sarda breed (Ss and St) had even negative values.

It is interesting to note that the Sarda breed showed significantly different scores between animals reared in Sardinia (Ss) (negative values) and those reared in Tuscany (St) (positive values) (Figure 1). This result could be linked to the adaptation of animals to different farming systems [42]. In fact, to our knowledge, there are no data showing strong genetic differences between Sarda sheep reared in the two regions. A summary of the findings obtained is shown in Table 3.

## 4. Conclusions

The MFA used in this work provide us with the possibility of studying the mammary metabolism of sheep through the reduction in a great number of variables to a few latent factors with biological meaning. This statistical approach grouped together FA involved in the same metabolic pathway, explaining the process of the secretion of fat in the mammary gland of sheep. The scores for these latent factors were consistently found to be influenced by ovine breeds. This approach proved to be an important tool for studying the effect of different sheep breeds in defining the FA profile of milk. In particular, it was possible to demonstrate that the breed also plays an important role in aspects of milk fat, which is believed to be influenced almost exclusively by the animal’s diet.

The MFA system makes it possible to identify traits on which to act in order to set up a future study of the genetic improvement of sheep species from the perspective of milk quality.

## Figures and Tables

**Figure 1 animals-12-00722-f001:**
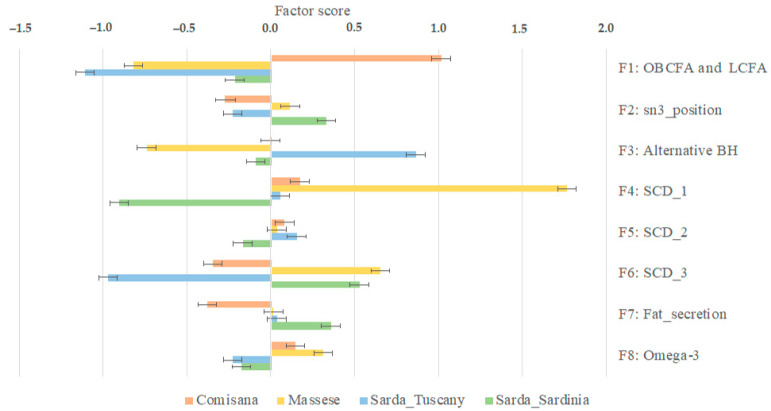
Effect of breed on the factors extracted by multivariate factor analysis. Legend: F1 *=* factor 1; F2 *=* factor 2; F3 *=* factor 3; F4 *=* factor 4; F5 *=* factor 5; F6 *=* factor 6; F7 *=* factor 7; F8 *=* factor 8; OBCFA *=* odd branched-chain fatty acid; BH *=* biohydrogenation; SCD *=* stearoyl-CoA desaturase.

**Table 1 animals-12-00722-t001:** Descriptive statistics for individual milk fatty acids (g/100 g of total lipids) (FA; *n* = 853).

	Mean	SD	CV%	P5	P95	Kurtosis
C4:0	3.24	0.81	25.03	2.32	4.90	0.95
C6:0	1.91	0.41	21.41	1.23	2.56	−0.14
C8:0	1.78	0.44	24.71	1.11	2.54	0.24
C10:0	5.52	1.63	29.54	3.05	8.45	0.36
C10:1c9	0.20	0.09	43.61	0.06	0.34	0.45
C12:0	3.28	0.91	27.69	2.07	4.91	0.91
C13:0	0.06	0.03	51.68	0.02	0.10	1.97
C14:0	9.70	1.79	18.51	7.25	12.78	0.43
C14:0iso	0.14	0.05	34.90	0.08	0.23	0.51
C14:1c9	0.24	0.15	63.27	0.09	0.61	1.58
C15:0	1.20	0.28	23.27	0.69	1.61	−0.21
C16:0iso	0.33	0.10	30.02	0.17	0.49	0.22
C16:0	23.11	3.42	14.78	17.52	28.92	0.05
C16:1c9	0.74	0.26	35.63	0.35	1.20	0.88
C18:0	9.82	2.08	21.13	6.77	13.24	0.81
C18:1t6–8	0.24	0.13	54.51	0.10	0.55	2.17
C18:1t9	0.28	0.11	39.36	0.16	0.52	2.04
C18:1t10	0.41	0.19	45.82	0.19	0.75	2.24
C18:1t11	2.33	1.45	62.08	0.83	4.74	2.89
C18:1c9	18.25	3.43	18.81	12.75	23.60	0.01
C18:1t15	0.43	0.16	36.70	0.17	0.70	0.16
C18:1c12	0.27	0.10	37.23	0.11	0.40	1.50
C18:2n6	2.18	0.55	24.99	1.30	3.05	0.32
C20:0	0.28	0.09	31.94	0.15	0.43	0.45
C18:3n3	1.10	0.49	44.54	0.39	2.09	0.96
C18:2c9t11	1.28	0.58	45.63	0.50	2.26	1.37
C20:1c11	0.04	0.02	57.62	0.01	0.08	1.90
C21:0	0.09	0.04	40.67	0.02	0.14	0.01
C20:4 n6	0.14	0.06	46.03	0.06	0.25	0.55
C20:5 n3	0.07	0.02	36.37	0.03	0.11	0.53
C24:0	0.06	0.03	42.39	0.03	0.11	0.95
C22:5 n3	0.14	0.05	37.85	0.07	0.23	0.47
C22:6 n3	0.06	0.04	66.26	0.00	0.14	0.91

SD = standard deviation; CV% = coefficient of variation; P5 = 5th percentile; P95 = 95th percentile.

**Table 2 animals-12-00722-t002:** Rotated factor (F) pattern and proposed factor name.

Name of Factors	OBCFA and LCFA	sn3_ Position	Alternative BH	SCD_1	SCD_2	SCD_3	Fat Secretion	Omega-3	
Factors	Factor 1	Factor 2	Factor 3	Factor 4	Factor 5	Factor 6	Factor 7	Factor 8	Communality
C13:0	**0.781**	−0.034	−0.036	0.177	−0.077	−0.059	−0.134	−0.145	0.692
C14:0iso	**0.877**	−0.172	−0.051	0.096	0.063	−0.080	0.147	−0.060	0.847
C15:0	**0.849**	0.030	−0.122	0.021	0.017	0.181	0.000	0.033	0.772
C16:0iso	**0.872**	−0.032	−0.098	−0.090	−0.021	0.085	0.150	−0.049	0.812
C20:0	**0.673**	−0.322	0.055	−0.076	−0.288	0.155	0.373	−0.009	0.813
C20:4 n6	**0.795**	−0.201	0.078	0.155	−0.237	−0.066	−0.183	0.241	0.854
C24:0	**0.686**	−0.073	−0.146	−0.235	−0.066	0.145	0.355	−0.008	0.734
C22:5 n3	**0.722**	−0.129	−0.046	−0.005	0.043	−0.115	−0.004	**0.690**	0.904
C22:6 n3	**0.714**	−0.277	0.040	0.016	−0.044	−0.227	−0.219	**0.669**	0.825
C8:0	−0.088	**0.754**	−0.142	0.239	−0.015	−0.350	−0.164	0.052	0.806
C10:0	−0.132	**0.942**	−0.164	0.040	−0.130	−0.063	−0.111	0.013	0.966
C12:0	−0.172	**0.916**	−0.132	−0.029	−0.140	0.176	−0.089	−0.001	0.945
C14:0	−0.182	**0.686**	−0.067	−0.326	−0.211	0.426	0.047	−0.037	0.845
C18:1c9	0.302	**−0.603**	0.173	0.214	−0.092	−0.006	0.056	0.295	0.629
C18:1t6–8	−0.218	−0.201	**0.814**	−0.115	0.321	−0.247	0.012	−0.105	0.938
C18:1t9	−0.282	−0.168	**0.771**	−0.109	0.408	−0.190	0.087	−0.044	0.927
C18:1t10	−0.097	−0.112	**0.809**	0.045	0.141	−0.103	−0.178	−0.085	0.748
C18:1c12	0.285	−0.151	**0.684**	−0.273	−0.145	0.065	0.006	−0.006	0.672
C18:1t15	−0.093	−0.081	**−0.663**	0.397	0.129	0.041	0.271	−0.163	0.731
C10:1c9	0.012	0.466	0.167	**0.638**	−0.035	0.045	−0.235	−0.110	0.723
C14:1c9	−0.145	0.085	−0.245	**0.637**	−0.048	0.468	−0.182	0.130	0.765
C18:1t11	−0.325	−0.082	0.240	−0.109	**0.822**	−0.211	0.053	0.025	0.906
C18:2c9t11	0.074	−0.153	0.168	−0.029	**0.910**	−0.071	−0.217	0.122	0.953
C16:0	0.028	0.162	−0.068	−0.335	−0.369	**0.680**	0.100	0.076	0.633
C16:1c9	0.120	−0.024	−0.191	−0.032	−0.082	**0.660**	−0.086	−0.127	0.518
C18:0	−0.119	−0.316	0.032	−0.084	−0.074	−0.204	**0.689**	0.034	0.517
C4:0	0.468	−0.178	−0.107	0.227	0.054	−0.314	**−0.663**	0.115	0.597
C6:0	0.158	0.180	−0.034	−0.216	0.018	−0.585	**−0.622**	−0.052	0.629
C20:5 n3	0.215	−0.028	−0.258	0.031	0.214	−0.074	0.005	**0.638**	0.573
C18:2n6	0.434	0.024	0.251	0.355	−0.118	0.016	0.147	0.286	0.495
C18:3n3	0.004	−0.094	0.121	−0.134	0.396	−0.410	0.158	0.214	0.437
C20:1c11	−0.365	0.043	0.171	0.279	0.180	0.001	0.004	−0.178	0.306
C21:0	0.302	−0.034	−0.123	−0.342	0.045	0.131	0.339	−0.042	0.361
Eigenvalue	7.32	5.76	3.84	2.99	2.09	1.55	1.37	1.22	
Variance explained	22.18	17.45	11.62	9.07	6.34	4.69	4.15	3.70	
Cumulative variance	22.18	39.63	51.26	60.33	66.66	71.36	75.50	79.20	

Values above 0.6 in bold.

**Table 3 animals-12-00722-t003:** Summary of findings regarding factors and breeds.

Factors	Findings
Factor 1 = OBCFA and LCFA	*Comisana* was the only breed that showed positive scores
Factor 2 = sn3_position	*Massese* and *Sarda_Sardinia* showed positive scores, while *Comisana* and *Sarda_Tuscany* showed negative scores
Factor 3 = Alternative BH	*Sarda_Tuscany* was the only breed that showed positive scores
Factor 4 = SCD_1	*Sarda_Sardinia* was the only breed that showed negative scores
Factor 5 = SCD_2	*Sarda_Sardinia* was the only breed that showed negative scores
Factor 6 = SCD_3	*Massese* and *Sarda_Sardinia* showed positive scores, while *Comisana* and *Sarda_Tuscany* showed negative scores
Factor 7 = Fat secretion	*Comisana* was the only breed that showed negative scores
Factor 8 = Omega-3	*Massese* and *Comisana* showed positive scores, while *Sarda_Sardinia* and *Sarda_Tuscany* showed negative scores

Legend: OBCFA *=* odd branched-chain fatty acid; BH *=* biohydrogenation; SCD *=* stearoyl-CoA desaturase.

## Data Availability

Not applicable.

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
