# Peer review of "Study of the Fatty Acid Profile of Milk in Different Sheep Breeds: Evaluation by Multivariate Factorial Analysis"

_animals, 2022, doi:10.3390/ani12060722_

Round 1

Author Response

Dear reviewer, we thank you for your suggestions, which certainly improved the quality of our manuscript.
We have tried to satisfy all your requests and suggestions. All changes are shown in red in the revised version of our manuscript.
Thanks for the work done.

The authors used the multivariate factor analysis (MFA) to investigate on the correlation patterns among sheep milk fatty acids, and tried to determine new synthetic variables for evaluate differences among breeds.

My major concern is about the poorly detailed feeding systems (pasture botanical composition, pasture dry matter intake, ingredients and chemical compostion of compound feed) as well as milk yield and days in milk.

AU: We provided all these information in a supplementary table (Table S1). Thank you.

 line 50: in sheep milk a high level of some short chain fatty acids (SCFA) is particularly referred to capric (C10:0) not to caproic (C6:0) and caprylic (C8:0) as confirmed also by the results reported in Table 1 of present trial;

AU: We agree with the reviewer even though in some circumstances even C6:0 and C8: 0 are quite high. However we have changed the text as suggested by the reviewer. (lane 51 in the revised manuscript)

line 101- 103: give informations on pasture and other parameters as mentioned above;

AU: We provided all these information in a supplementary table (Table S1). Thank you.

line 190: to better standardize factors other than breed, the authors have to report detailed

AU: Done as suggested by the reviewer (lane 194-195 in the revised manuscript)

informations on feeeding systems;

AU: We provided more information about feeding system (see lanes 104-118)

Table 1: give informations on how the results of different FAs are expressed

AU: we provided how FA are expressed.

Reviewer 2 Report

Introduction

This is OK in general, although it would benefit from a reduction in length. However, I leave this to the authors.

The hypothesis of the authors must be clearly presented.

Procedures.

The sampling procedure is described inadequately. The description should be extended to include more details to allow a full evaluation of the procedure.

Results & discussion

Please present a table similar to Table 1 separate for each breed as supplementary material, i.e. four different tables.

Please add legend in Figure 1.

Please colourise Figure 1

Please add a new table with a summary of the findings per breed and factor. The text is lengthy and complicated, therefore a table will help readers for easy evaluation of findings.

Some recent relevant references are missing and should be added.

Author Response

We thank you for your suggestions, which certainly improved the quality of our manuscript.
We have tried to satisfy all your requests and suggestions. All changes are shown in red in the revised version of our manuscript.
Thanks for the work done.

Introduction

This is OK in general, although it would benefit from a reduction in length. However, I leave this to the authors.

AU: We understand the reviewer's comment. This type of statistical analysis involves having to consider several aspects at the same time. The consequence is a structure of the manuscript that must take into account and explain the various relationships that exist between the variables considered and the factors. it is very difficult to simplify the text without losing important information. We believe that for the completeness of the work it is better not to modify the text. We have tried by adding tables, as suggested by the reviewer himself, to provide an easier reading for the reader.

The hypothesis of the authors must be clearly presented.

AU: We tried to clear our hypothesis. See lane 86-89 in the revised manuscript.

Procedures.

The sampling procedure is described inadequately. The description should be extended to include more details to allow a full evaluation of the procedure.

AU: We improve the sampling procedure description. See lanes 114-118 in the revised manuscript

Results & discussion

Please present a table similar to Table 1 separate for each breed as supplementary material, i.e. four different tables.

AU: Done as suggested by the reviewer. Thank you

Please add legend in Figure 1.

AU: Done. Thank you for the suggestion.

Please colourise Figure 1

AU: Done. Thank you for the suggestion

Please add a new table with a summary of the findings per breed and factor. The text is lengthy and complicated, therefore a table will help readers for easy evaluation of findings.

AU: Done. Thank you for the suggestion

Some recent relevant references are missing and should be added.

AU: Done.

Round 2

Reviewer 1 Report

The authors addressed all the reviewers suggestions, highly improving their manuscript.  

Author Response

Thank you for your comments and suggestions.

Reviewer 2 Report

The manuscript has been improved.
Can the authors please check one more time the equations in subsection 2.3. to confirm that they are correct, please?

The academic editor can confirm the corrections. No need to return to this reviewer.

After that, the manuscript is ready for acceptance.

Author Response

We confirm that the equations  in the section 2.3 are correct.